# Crystal structure of a natural light-gated anion channelrhodopsin

Hai Li[1], Chia-Ying Huang[2], Elena G Govorunova[1], Christopher T Schafer[1], Oleg A Sineshchekov[1], Meitian Wang[2], Lei Zheng[1]*, John L Spudich[1]*

[1]Department of Biochemistry and Molecular Biology, Center for Membrane Biology, University of Texas Health Science Center – McGovern Medical School, Houston, United States; [2]Swiss Light Source, Paul Scherrer Institute, Villigen, Switzerland

**Abstract** The anion channelrhodopsin *Gt*ACR1 from the alga *Guillardia theta* is a potent neuron-inhibiting optogenetics tool. Presented here, its X-ray structure at 2.9 Å reveals a tunnel traversing the protein from its extracellular surface to a large cytoplasmic cavity. The tunnel is lined primarily by small polar and aliphatic residues essential for anion conductance. A disulfide-immobilized extracellular cap facilitates channel closing and the ion path is blocked mid-membrane by its photoactive retinylidene chromophore and further by a cytoplasmic side constriction. The structure also reveals a novel photoactive site configuration that maintains the retinylidene Schiff base protonated when the channel is open. These findings suggest a new channelrhodopsin mechanism, in which the Schiff base not only controls gating, but also serves as a direct mediator for anion flux.
DOI: https://doi.org/10.7554/eLife.41741.001

## Introduction

Anion channelrhodopsins (ACRs) are natural light-gated anion channels first discovered in the cryptophyte alga *Guillardia theta* (*Gt*ACR1 and *Gt*ACR2) (*Govorunova et al., 2015*). Their large Cl⁻ conductance makes *Gt*ACRs and other ACRs later found in various cryptophyte species (*Govorunova et al., 2018*; *Govorunova et al., 2017b*) the most potent neuron-silencing optogenetic tools available. *Gt*ACRs have proven to be effective inhibitors of neural processes and behavior in flies (*Mauss et al., 2017*; *Mohammad et al., 2017*; *Steck et al., 2018*), worms (*Bergs et al., 2018*), zebrafish (*Mohamed et al., 2017*), ferrets (*Wilson et al., 2018*), and mice (*Forli et al., 2018*; *Mahn et al., 2018*; *Messier et al., 2018*; *Wei et al., 2018*).

Of the 35 ACR homologs found in cryptophyte algae (*Govorunova et al., 2018*; *Govorunova et al., 2016*; *Wietek et al., 2016*), *Gt*ACR1 is the best characterized in terms of its gating mechanism and photochemical reaction cycle (*Sineshchekov et al., 2015*; *Sineshchekov et al., 2016*), and also is the only ACR for which light-gated anion conductance has been proven to be maintained in vitro in a purified state (*Li et al., 2016*) further recommending it as the preferred ACR for crystallization. The effects of mutations of several key residues, including E68Q/R, S97E, C102A and D234N, on photocurrents and photochemical conversions of *Gt*ACR1 have been studied in detail (*Sineshchekov et al., 2015*; *Sineshchekov et al., 2016*).

The most closely related molecules to ACRs are cation channelrhodopsins (CCRs) from chlorophyte algae (*Govorunova et al., 2017a*). The best characterized CCRs are channelrhodopsin-2 (*Cr*ChR2) (*Nagel et al., 2003*), a membrane-depolarizing phototaxis receptor from *Chlamydomonas reinhardtii* (*Sineshchekov et al., 2002*), and C1C2, a chimera of *Cr*ChR2 and its paralog *Cr*ChR1 (*Kato et al., 2012*). Atomic structures of C1C2 and *Cr*ChR2 have been obtained by X-ray crystallography (*Kato et al., 2012*; *Volkov et al., 2017*).

The two channelrhodopsin families exhibit large differences in their sequences and photochemistry (*Govorunova et al., 2017a*): (i) ACRs conduct only anions with complete exclusion of cations,

*For correspondence:
Lei.Zheng@uth.tmc.edu (LZ);
John.L.Spudich@uth.tmc.edu (JLS)

even $H^+$, for which CCRs exhibit their highest relative permeability; (ii) ACRs are generally more potent; for example $Gt$ACR1 exhibits 25-fold higher unitary conductance than $Cr$ChR2; (iii) The retinylidene Schiff base in the photoactive site deprotonates prior to channel opening in CCRs (*Lórenz-Fonfría and Heberle, 2014*) and, in contrast, in ACRs remains protonated throughout the lifetime of the open-channel state with deprotonation correlated with the initial phase of channel closing (*Sineshchekov et al., 2016*; *Wietek et al., 2016*).

Here, we report the atomic structure of the dark (closed) state of $Gt$ACR1, which is essential for elucidating the mechanism of its unique natural function of light-gated anion conductance through biological membranes. Also, understanding ACR mechanisms at the atomic scale would enable rational engineering to tailor their use as optogenetic tools.

## Results and discussion

### Overall $Gt$ACR1 structure

The $Gt$ACR1 protein was expressed in insect (Sf9) cells and purified as a disulfide-crosslinked homodimer (*Figure 1—figure supplement 1*). We obtained lipidic cubic phase (LCP) crystals of $Gt$ACR1, applied the continuous grid-scan method (*Wojdyla et al., 2016*) for X-ray data collection, and determined the structure at 2.9 Å resolution using molecular replacement (*Figure 1*, *Table 1*). Each asymmetric unit contains a $Gt$ACR1 homodimer molecule (*Figure 1—figure supplement 2*). Each monomer is composed of an extracellular cap domain, seven transmembrane helices (TM1-7), and a cytoplasmic loop at the carboxyl-terminus (*Figure 1*). In the extracellular domain, two kinked α-helices from the amino-terminal fragment and a β-hairpin from the TM2-3 loop lay on the interface of the membrane domain. The $Gt$ACR1 homodimer is stabilized by TM3 and TM4 interactions between monomers and further by an intermolecular disulfide bridge formed by the C6 residues (*Figure 1A–B*). Since TM5-7 are much longer than TM1-4, this dimeric arrangement creates a large funnel-shaped cytoplasmic cavity (~18 Å deep and ~28 Å wide). Despite the modest ~24% amino acid sequence identity between $Gt$ACR1 and C1C2/$Cr$ChR2, the structure of each $Gt$ACR1 protomer can be superposed well (*Figure 1—figure supplement 3*) with either of the two CCR structures (r.m.s.d. 0.9 Å) indicating that these functionally distinct channelrhodopsins share a common TM helical scaffold conformation in their closed states.

### The anion conductance pathway

A continuous tunnel spanning through the protein from the extracellular to cytoplasmic surface was detected in each $Gt$ACR1 protomer by serial cross-sections (*Figure 2A*). The tunnel, assembled by TM1-3 and 7, starts from an electropositive port on the extracellular surface, intersects the retinylidene Schiff base in the middle of the membrane, and ends at an intracellular port deeply embedded in the large dimeric cavity (*Figure 2B*).

The continuous intramolecular tunnel in $Gt$ACR1 directly visualized by cross-section, presumably indicating the anion conductance pathway, was also detected by the program CAVER (probe radius 0.9 Å) (*Kozlikova et al., 2014*) (*Figure 2—figure supplement 1A*). For comparison, only a partial tunnel open on the extracellular side was found in C1C2 (*Kato et al., 2012*) (*Figure 2—figure supplement 1B*), and we found no tunnel open to either surface with CAVER in $Cr$ChR2.

Despite the high similarity of the TM helix scaffolds of $Gt$ACR1 and C1C2/$Cr$ChR2, the tunnel of $Gt$ACR1 is primarily lined by small polar and aliphatic residues (*Figure 2C*) in contrast to charged residues in the corresponding positions in C1C2 and $Cr$ChR2: A75 vs E136/E97 (C1C2/$Cr$ChR2 numbering), T71 vs K132/K93; S97 vs E162/E123, A61 vs E122/E83, and L108 vs H173/H134 (*Figure 2—figure supplement 2* top row). Tunnel-lining residues also include R94 (R159/R120) and D234 (D292/D253) (*Figure 2—figure supplement 2*, bottom row), highly conserved in the photoactive sites of microbial rhodopsins, and E68 (E129/E90), characteristic of both ACRs and chlorophyte CCRs. The differences in $Gt$ACR1 from the CCR structures significantly reduce the negativity of the putative channel pore lining consistent with anion vs. cation conductance.

### The extracellular port cap

A unique structural feature is found in the extracellular domain of $Gt$ACR1. In addition to the disulfide link between the two protomers, an intraprotomer disulfide bridge is formed between C21 from

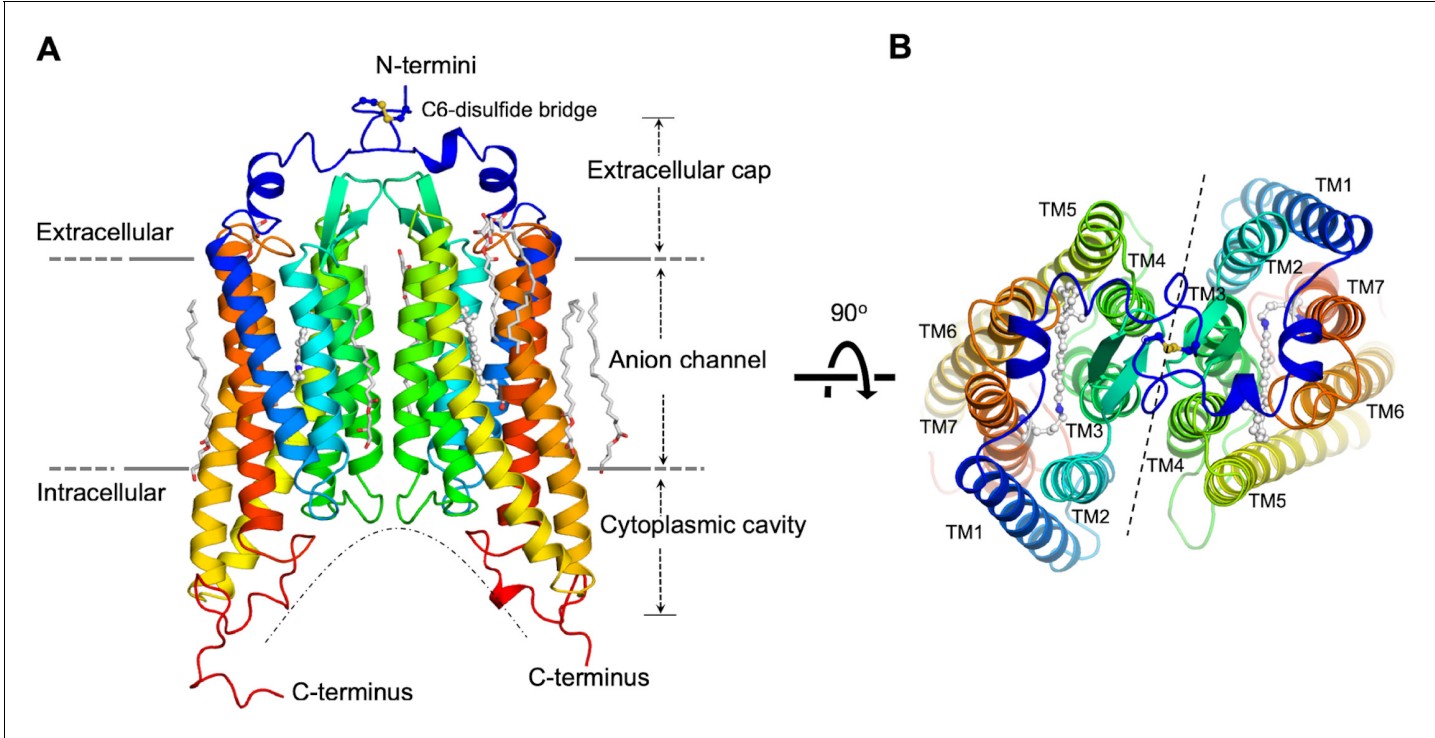

**Figure 1.** Overall structure of the *Gt*ACR1 homodimer. Side (**A**), and top (**B**) views. Each *Gt*ACR1 protomer is depicted in cartoon with the N-termini in *blue* and the C-termini in *red*. Retinal prosthetic groups (stick-balls) are embedded in the 7TMs. An intermolecular disulfide bridge is formed by C6 (*yellow* sticks). Resolved monoolein lipids are shown as sticks.
DOI: https://doi.org/10.7554/eLife.41741.002

The following figure supplements are available for figure 1:

**Figure supplement 1.** Structural determination of *Gt*ACR1.
DOI: https://doi.org/10.7554/eLife.41741.003

**Figure supplement 2.** Crystal packing of the *Gt*ACR1 structure.
DOI: https://doi.org/10.7554/eLife.41741.004

**Figure supplement 3.** Conserved 7-TM conformation of *Gt*ACR1.
DOI: https://doi.org/10.7554/eLife.41741.005

the amino-terminal segment and C219 within the TM6-7 loop (*Figure 3A*). This intramolecular cross-link immobilizes the kinked helices to the retinal-conjugated TM7, and encaps a hydrophobic part of the segment on the extracellular tunnel entry port (*Figure 3B*). Disrupting this extracellular loop conformation, either by truncation of the amino-terminal loop (Δ1–25) or by substituting C21 and C219 with serine to abolish the intramolecular disulfide, resulted in slowed channel closing (*Figure 3C*). Both C21 and C219 are highly conserved in ACRs (*Govorunova et al., 2017b*), but not in CCRs, revealing a role of this intramolecular disulfide bridge specific to the ACR family.

## Ion pathway constrictions

The intramolecular tunnel in *Gt*ACR1 is constricted at three positions: at the extracellular port (C1), near the photoactive retinylidene Schiff base (C2), and at the cytoplasmic side (C3) (*Figures 2D* and *3D–F*). Near the extracellular port, the C1 constriction (*Figure 3D*) is stabilized by an H-bond network adjacent to the disulfide-immobilized extracellular cap and formed by the side chains of Y81, R94 and E223 (*Figure 3A*). The mutation R94A nearly abolished Cl⁻ conductance (Figure 5D). To analyze the influence of mutations on channel kinetics, we used laser flash excitation (single-turnover conditions), because under continuous light a mixture of intermediates is formed, the composition of which depends on the intensity and duration of illumination that may influence the kinetics. Channel closing in the wild-type *Gt*ACR1 is biphasic (*Sineshchekov et al., 2015*). As shown in *Figure 3—figure supplement 1* and reported earlier for the E223Q mutant (*Sineshchekov et al., 2015*), all three

**Table 1.** Crystallographic data and refinement statistics of the *Gt*ACR1 structure

| PDB ID | 6EDQ |
| --- | --- |
| Space group | $P2_12_12$ |
| a, b, c (Å) | 77.79, 149.55, 62.41 |
| α, β. γ (°) | 90, 90, 90 |
| Beamline | SLS-X06SA |
| Wavelength (Å) | 1.0 |
| Resolution (Å) | 47.91–2.9 (2.98–2.9) |
| *R*meas | 0.39 (2.65) |
| I /σ (I) | 4.54 (0.97) |
| Completeness (%) | 99.8 (100) |
| Multiplicity | 6.89 (6.57) |
| CC1/2 (%) | 99 (32) |
| Refinement | |
| Resolution (Å) | 47.91–2.90 (3.08-2.90) |
| No. of unique reflections | 16711(2732) |
| *R*work/*R*free | 0.23/0.27 |
| R.m.s. deviations | |
| Bond lengths (Å) | 0.005 |
| Bond angles (°) | 0.892 |
| B-factor | |
| Proteins | 63.8 |
| Ligands | 80.1 |
| $H_2O$ | 51.0 |
| Ramachandran Plot | |
| Favored (%) | 97.56 |
| Allowed (%) | 2.44 |
| MolProbity Clash score | 9.39 |

*Data processing statistics are reported with Friedel pairs merged. Values in parentheses are for the highest resolution shell.

DOI: https://doi.org/10.7554/eLife.41741.006

mutations strongly slowed the slow decay phase to a similar extent as that observed in the C21S_C219S mutant. These results suggest that the combination of the H-bond network of E223 and its neighbouring intraprotomer disulfide bridge controls the rate of channel closing in the extracellular region and stabilizes the essential residue R94 in the closed state.

The narrowest constriction C2 lies at the photoactive site and is formed by the side chains of T101, L64, and M105 (*Figure 3E*). Four of the five residues that form the intracellular constriction C3 (L108, A61, E60, L245 and P58) (*Figure 3F*) are in corresponding positions as the residues that form the 'intracellular gate' in CCRs (*Deisseroth and Hegemann, 2017*), but in *Gt*ACR1 and other ACRs only E60 (E121/E82) is shared with CCRs. The *Gt*ACR1 structure that we obtained from dark-grown crystals is presumably the dark (closed) state of the channel protein. To examine the role of these contriction-forming residues in the channel open state, we scanned the tunnel constrictions with Glu substitutions and measured photocurrents in the respective mutants. We chose Glu as a substituent because two of the constriction residues, A61 and A75, correspond to the highly conserved Glu residues in CCRs (E122/E83 and E136/E97 in C1C2/*Cr*ChR2, respectively), and because neutralization of E83 was required for elimination of the residual $H^+$ permeability in Cl⁻-conducting CCR mutants (*Berndt et al., 2016*; *Wietek et al., 2015*). We also hypothesized that the bulky negatively charged Glu side chain would block the *Gt*ACR1 channel when placed in the ion conduction pathway. Indeed,

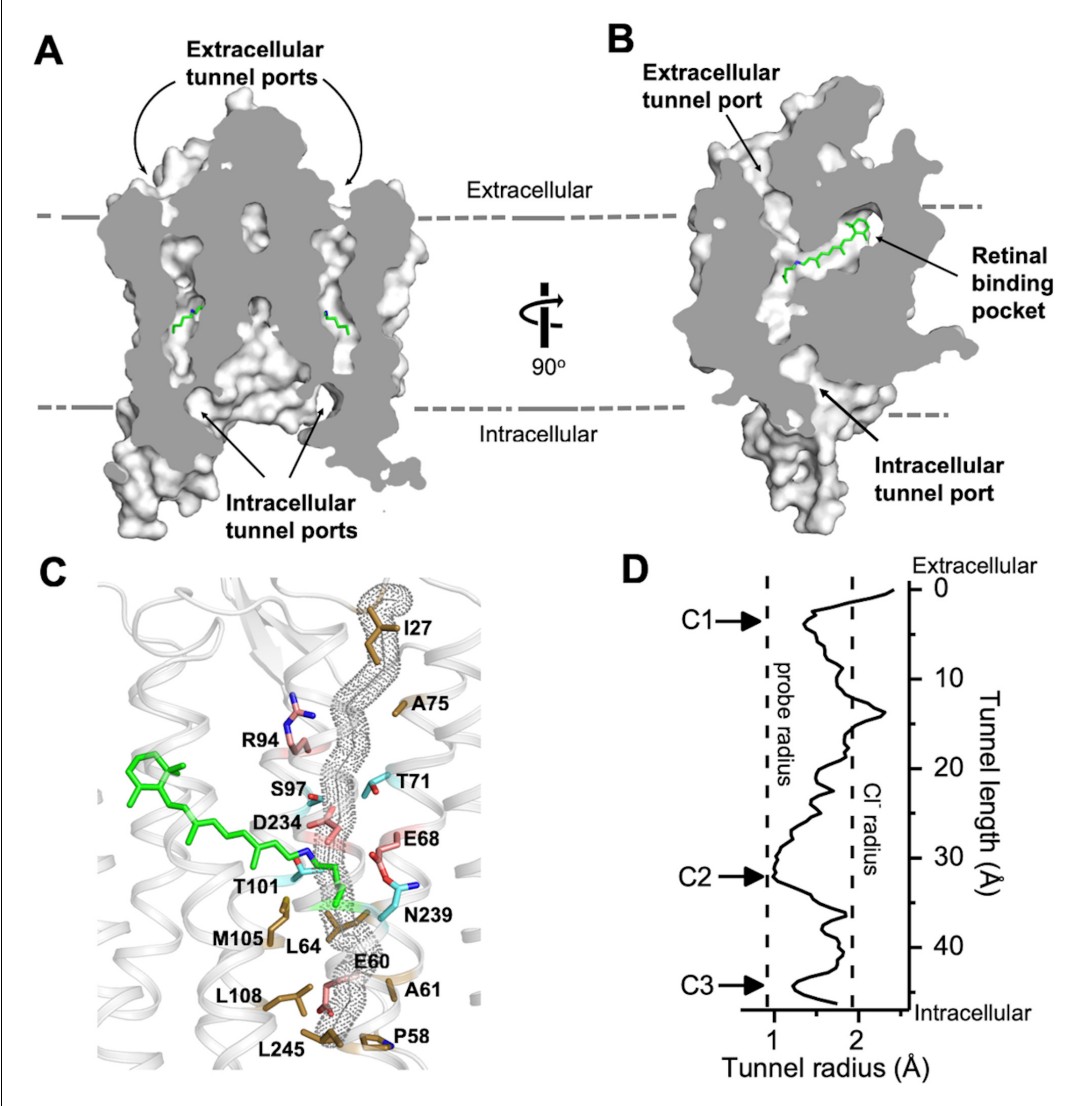

**Figure 2.** The dark state tunnel of *Gt*ACR1. (**A**) A cross-section view of the *Gt*ACR1 dimer showing two continuous intramolecular tunnels traversing from extracellular ports to the cytoplasmic cavity; retinal (*green*). (**B**) A cross-section view of a *Gt*ACR1 protomer showing the conformation of the transmembrane ion tunnel and retinal binding pocket connected at the retinylidene Schiff base position. (**C**) The tunnel (dots) detected by CAVER with tunnel-lining residues (sticks): charged (*red*), polar (*cyan*), and apolar residues (*clay*). (**D**) The tunnel profile of *Gt*ACR1 detected by CAVER; the arrows indicate three constrictions C1-C3.

DOI: https://doi.org/10.7554/eLife.41741.007

The following figure supplements are available for figure 2:

**Figure supplement 1.** The predicted tunnel path of *Gt*ACR1.

DOI: https://doi.org/10.7554/eLife.41741.008

**Figure supplement 2.** Comparison of selected tunnel-lining residues in *Gt*ACR1 with their counterparts in C1C2 and *Cr*ChR2.

DOI: https://doi.org/10.7554/eLife.41741.009

perturbation of any residues at C2 or C3 greatly reduced or eliminated the photocurrents, while effects of most mutations (except A75E) at the C1 position were negligible (*Figure 3G*), suggesting that in the open conformation the channel is wider in the extracellular portion and more narrow in its central and intracellular stretches. Kinetically, the mutations of the C1 and C3 residues mostly affected the slow phase of channel closing, making it slower than that in the wild-type (*Figure 3— figure supplement 2*). Accurate kinetic analysis of the C2 mutations was not possible because of their greatly reduced photocurrents.

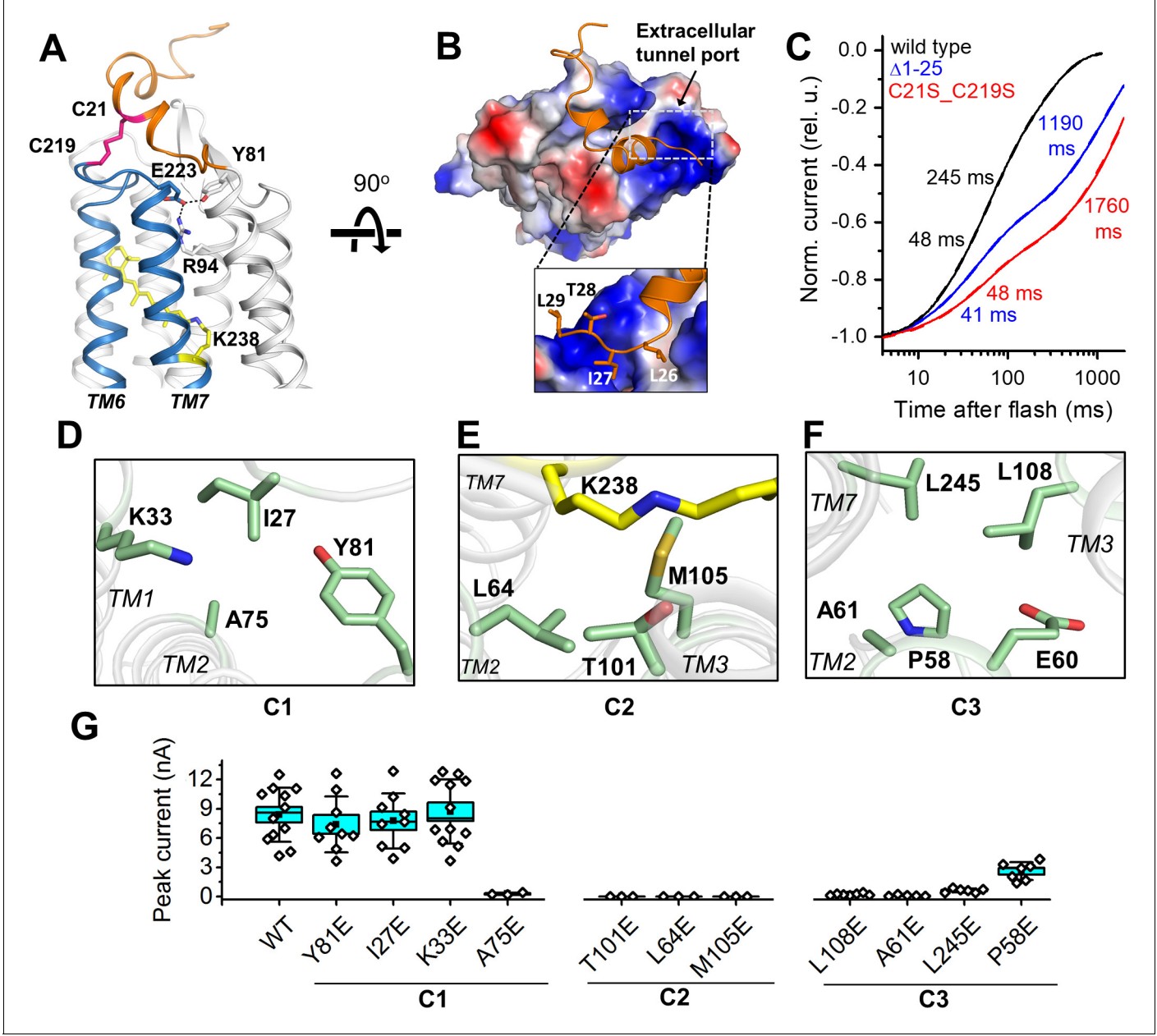

**Figure 3.** Features of the ion pathway of *Gt*ACR1. (A) The N-terminal extracellular loop (*orange*) immobilized by an intracellular C21-C219 disulfide bridge (*red*) to the TM6-7 loop (*blue*); an H-bond network (*black* dashed lines) formed by residues (sticks) near the extracellular port. (B) The hydrophobic segment (*orange*) blocks the extracellular port rendered by the electrostatic potential surface. Rectangle: closer (rotated) view of the peptide cap conformation. (C) Decay kinetics of laser flash-evoked photocurrents by the wild-type *Gt*ACR1 and indicated mutants. (D–F) The structure of the three constrictions: C1 (D), C2 (E), and C3 (F). (G) Peak photocurrents generated by Glu substitutions of the constriction residues in response to a 1 s light pulse (515 nm, 7.7 mW mm$^{-2}$) with 131 mM Cl$^-$ in the pipette and 6 mM Cl$^-$ in the bath. The black squares, mean; line, median; box, SE; whiskers, SD; empty diamonds, raw data recorded from individual cells.

DOI: https://doi.org/10.7554/eLife.41741.010

The following source data and figure supplements are available for figure 3:

**Source data 1.** Numerical data for the current amplitude values measured in individual cells are shown in *Figure 3G*.
DOI: https://doi.org/10.7554/eLife.41741.020
**Source data 2.** Numerical data for the reversal potential values measured in individual cells are shown in *Figure 3—figure supplements 4* and *6*.
DOI: https://doi.org/10.7554/eLife.41741.021
**Source data 3.** Numerical data for the reversal potential values measured in individual cells are shown in *Figure 3—figure supplement 9A*.
DOI: https://doi.org/10.7554/eLife.41741.022

*Figure 3 continued on next page*

*Figure 3 continued*

**Source data 4.** Numerical data for the reversal potential values measured in individual cells are shown in *Figure 3—figure supplement 9B*.
DOI: https://doi.org/10.7554/eLife.41741.023

**Figure supplement 1.** Laser flash-evoked photocurrents generated by the *Gt*ACR1_Y81F and R94A mutants, as compared to the wild-type.
DOI: https://doi.org/10.7554/eLife.41741.011

**Figure supplement 2.** Laser flash-evoked photocurrents generated by the mutants of the C1 and C3 residues.
DOI: https://doi.org/10.7554/eLife.41741.012

**Figure supplement 3.** The current-voltage relationships of the the mutants of the C1 and C3 residues measured under a $Cl^-$ gradient.
DOI: https://doi.org/10.7554/eLife.41741.013

**Figure supplement 4.** The reversal potentials in the mutants of the C1 and C3 residues determined under a $Cl^-$ gradient.
DOI: https://doi.org/10.7554/eLife.41741.014

**Figure supplement 5.** The current-voltage relationships of the mutants of Q46 and positively charged residues measured under a $Cl^-$ gradient.
DOI: https://doi.org/10.7554/eLife.41741.015

**Figure supplement 6.** The reversal potentials in the mutants of Q46 and positively charged residues determined under a $Cl^-$ gradient.
DOI: https://doi.org/10.7554/eLife.41741.016

**Figure supplement 7.** The current-voltage relationships of the mutants of Q46 and positively charged residues measured under a $H^+$ gradient.
DOI: https://doi.org/10.7554/eLife.41741.017

**Figure supplement 8.** The current–voltage relationships of the mutants of Q46 and positively charged residues measured under a $Na^+$ gradient.
DOI: https://doi.org/10.7554/eLife.41741.018

**Figure supplement 9.** The reversal potentials in the mutants of Q46 and positively charged residues determined under $H^+$ and $Na^+$ gradients.
DOI: https://doi.org/10.7554/eLife.41741.019

To test whether the mutations of the constriction residues compromised permeability of the channel for $Cl^-$, we partially replaced this ion in the bath with non-permeable aspartate (*Govorunova et al., 2015*) and measured the current-voltage relationships (IE curves) (*Figure 3—figure supplement 3*) to determine the reversal potentials ($E_{rev}$). None of the mutants in which photocurrents near the reversal potential could be resolved from the background noise yielded $E_{rev}$ values that were statistically significantly different from that measured in the wild type (*Figure 3—figure supplement 4*; full statistical results are in *Table 2*). Several other mutations were recently reported to shift the $E_{rev}$ under $Cl^-$ gradient conditions from the $Cl^-$ Nernst potential (*Kato et al., 2018*; *Kim et al., 2018*). However, we found no such change when we tested seven of these single and double mutants (*Figure 3—figure supplements 5* and *6*; full statistical results are in *Table 2*). The shifts were attributed to disrupting anion selectivity in *Gt*ACR1 by the mutations resulting in cation permeability (*Kato et al., 2018*; *Kim et al., 2018*). Therefore, we further tested these mutants in the presence of $H^+$ and $Na^+$ gradients (*Figure 3—figure supplements 7* and *8*, respectively) and again found no statistically significant difference from the wild type (*Figure 3—figure supplement 9*; full statistical results are in *Tables 3* and *4*), indicating that none of these mutations produced permeability for these cations in *Gt*ACR1.

## The retinylidene Schiff base

Confocal near-infrared resonance Raman spectroscopy has shown that the unphotolyzed state of *Gt*ACR1 contains almost exclusively all-*trans* retinal (*Yi et al., 2016*). In the middle of the protein, all-*trans*-retinal covalently bound by a Schiff base linkage to K238 is found in an elongated cavity formed by conserved hydrophobic residues. While the conformations of the retinal polyene chain are nearly identical in *Gt*ACR1 and C1C2/*Cr*ChR2, the presence of F160 in *Gt*ACR1 (G224/G185 in C1C2/*Cr*ChR2, respectively) pushes the β-ionone ring towards the extracellular side by 1.2 Å (*Figure 4*). Despite this conformational difference, the action spectrum of photocurrents generated by the F160G mutant was almost identical to that of the wild-type *Gt*ACR1 (*Figure 4—figure supplement 1*).

Remarkable structural differences between *Gt*ACR1 and the two crystallized CCRs are found in the retinylidene Schiff base environment. In C1C2 and *Cr*ChR2 the protonated Schiff base (PSB) participates in a quadruple salt-bridge network formed with D292/D253, E162/E123 and K132/K93 side-chains (*Figure 5B*). However, this strong network is absent in the *Gt*ACR1 structure due to the replacement of E162/E123 and K132/K93 with smaller uncharged residues S97 and T71, respectively (*Figure 5A*). D234 is the only residue directly interacting with the protonated Schiff base (PSB) in

**Table 2.** The output of Kriskal-Wallis ANOVA analysis of the results shown in *Figure 3—figure supplements 4* and *6*

| X-Function | | | Kruskal-Wallis ANOVA | | | |
|---|---|---|---|---|---|---|
| Data filter | | | No | | | |
| Variant | | Data | | | Range (number of cells) | |
| WT | | [DataAsp]Sheet1!WT | | | [1*:10*] | |
| Y81E | | [DataAsp]Sheet1!Y81E | | | [1*:7*] | |
| I27E | | [DataAsp]Sheet1!I27E | | | [1*:7*] | |
| K33E | | [DataAsp]Sheet1!K33E | | | [1*:8*] | |
| L108E | | [DataAsp]Sheet1!L108E | | | [1*:7*] | |
| A61E | | [DataAsp]Sheet1!A61E | | | [1*:5*] | |
| L245E | | [DataAsp]Sheet1!L245E | | | [1*:8*] | |
| P58E | | [DataAsp]Sheet1!P58E | | | [1*:7*] | |
| Q46A | | [DataAsp]Sheet1!Q46A | | | [1*:13*] | |
| K188A | | [DataAsp]Sheet1!K188A | | | [1*:8*] | |
| K188E | | [DataAsp]Sheet1!K188E | | | [1*:7*] | |
| R192E | | [DataAsp]Sheet1!R192E | | | [1*:9*] | |
| Q46AK188A | | [DataAsp]Sheet1!Q46AK188A | | | [1*:8*] | |
| K188AR192A | | [DataAsp]Sheet1!K188AR192A | | | [1*:7*] | |
| K188AR259A | | [DataAsp]Sheet1!K188AR259A | | | [1*:7*] | |
| Variant | N | Min | Q1 | Median | Q3 | Max |
| WT | 10 | 61 | 70 | 75.5 | 90.25 | 93 |
| Y81E | 7 | 71 | 73 | 77 | 85 | 85 |
| I27E | 7 | 75 | 77 | 84 | 86 | 86 |
| K33E | 8 | 70 | 73.75 | 80 | 85.75 | 87 |
| L108E | 7 | 69 | 72 | 75 | 76 | 82 |
| A61E | 5 | 69 | 71 | 74 | 80 | 80 |
| L245E | 8 | 71 | 73.25 | 75.5 | 85 | 91 |
| P58E | 7 | 63 | 70 | 75 | 84 | 89 |
| Q46A | 13 | 69 | 74 | 83 | 90.5 | 96 |
| K188A | 8 | 78 | 78 | 80 | 82.5 | 83 |
| K188E | 7 | 75 | 76 | 79 | 81 | 83 |
| R192E | 9 | 75 | 78.5 | 80 | 81.5 | 86 |
| Q46AK188A | 8 | 73 | 76.75 | 79 | 82.25 | 84 |
| K188AR192A | 7 | 68 | 74 | 82 | 84 | 88 |
| K188AR259A | 7 | 76 | 79 | 84 | 90 | 92 |
| Variant | N | | Mean Rank | | Sum Rank | |
| WT | 10 | | 49.55 | | 495.5 | |
| Y81E | 7 | | 55.71429 | | 390 | |
| I27E | 7 | | 74.28571 | | 520 | |
| K33E | 8 | | 62.8125 | | 502.5 | |
| L108E | 7 | | 31 | | 217 | |
| A61E | 5 | | 36.4 | | 182 | |
| L245E | 8 | | 50.125 | | 401 | |
| P58E | 7 | | 47.64286 | | 333.5 | |
| Q46A | 13 | | 70.30769 | | 914 | |
| K188A | 8 | | 65.9375 | | 527.5 | |

*Table 2 continued on next page*

*Table 2 continued*

| X-Function | | Kruskal-Wallis ANOVA | |
| --- | --- | --- | --- |
| K188E | 7 | 58.14286 | 407 |
| R192E | 9 | 66.72222 | 600.5 |
| Q46AK188A | 8 | 59.25 | 474 |
| K188AR192A | 7 | 66.5 | 465.5 |
| K188AR259A | 7 | 84.42857 | 591 |
| Chi-Square | DF | Prob > Chi-Square | |
| 16.94395 | 14 | 0.25918 | |

Null Hypothesis: The samples come from the same population.

Alternative Hypothesis: The samples come from different populations.

At the 0.05 level, the populations are NOT significantly different.

DOI: https://doi.org/10.7554/eLife.41741.024

*Gt*ACR1, and its electrostatic interaction appears to be weakened by two H-bonds from tyrosine residues Y72 and Y207 (*Figure 5A*). The proton pump bacteriorhodopsin exhibits similar tyrosinyl H-bond-weakened interactions of D212, the residue in the corresponding position as D234. The interactions prevent D212 from accepting the Schiff base proton, which is transferred instead to D85 in the proton release pathway (*Luecke et al., 1998*). Resonance Raman and UV-vis absorption spectra of the D234N mutant of *Gt*ACR1 indicate that D234 is similarly neutral and not a Schiff base proton acceptor (*Sineshchekov et al., 2016*; *Yi et al., 2016*). The dark structure therefore may explain the persistence of protonation of the Schiff base throughout the lifetime of the open channel conformation in *Gt*ACR1. Of the two tyrosine residues, Y207 appeared to be more important functionally, as its replacement with phenylalanine suppressed the photocurrents to a greater extent than that of Y72 (*Figure 5D*) and caused a 12 nm blue shift of the action spectrum (*Figure 5—figure supplement 1*). The Y72F mutation mostly affected the slow decay phase, while the Y207F mutation caused a strong perturbation of both phases (*Figure 5—figure supplement 2*).

In *Cr*ChR2, photoisomerization of the Schiff base rapidly disrupts the strong salt-bridged network by inducing transfer of the Schiff base proton to D253 or E123 in ~10 µs prior to channel opening (*Lórenz-Fonfría and Heberle, 2014*). In contrast, the Schiff base remains protonated throughout the lifetime of the open channel conformation in *Gt*ACR1, and deprotonation of the Schiff base proton occurs late in the photocycle (~20 ms) correlated with fast channel closing (*Sineshchekov et al., 2016*). Unlike in the salt-bridge network around the Schiff base in the CCRs (*Figure 5B*), in *Gt*ACR1 no immediate proton accepting residue is available in the vicinity of the PSB and therefore later structural changes are required to enable Schiff base proton transfer, possibly to E68 (*Figure 5A*).

The location of the PSB, centered within the anion path in *Gt*ACR1, suggests that it may play a direct role in anion translocation in the open channel state. Consistent with this idea is that the PSB is only partially neutralized by its D234 counterion because the counterion is weakened by its interaction with the two tyrosinyl residues. Therefore, the PSB presents a partial positive charge capable of transient reversible interaction with Cl⁻ ions in a channel that is largely lined by small polar and hydrophobic aliphatic residues. Supporting a possible direct role of the PSB in the channel's permeability for anions, late deprotonation of the Schiff base after channel opening occurs in all three ACRs so far examined: *Gt*ACR1 and *Gt*ACR2 (*Sineshchekov et al., 2016*) and *Psu*ACR1 (*Wietek et al., 2016*), yet Schiff base deprotonation after channel opening is not known to occur in any CCR. Further indicating an essential role of the protonated Schiff base form, the mutant S97E, in which a potential Schiff base proton acceptor is placed at the corresponding position in *Gt*ACR1 as in CCRs and many other microbial rhodopsins, exhibits (i) appearance of fast Schiff base deprotonation, and (ii) a > 30 fold suppression of the amplitude of the chloride photocurrent (*Sineshchekov et al., 2016*). Furthermore, the double mutation Y207F/Y72F, expected to release inhibition of D234 as a proton acceptor, decreased the photocurrent amplitude to a negligible value (*Figure 5D*).

**Table 3.** The output of Kriskal-Wallis ANOVA analysis of the results shown in *Figure 3—figure supplement 9A*

| X-Function | | | | Kruskal-Wallis ANOVA | | |
|---|---|---|---|---|---|---|
| Data filter | | | | No | | |
| Variant | | | Data | | Range (number of cells) | |
| WT | | | [DatapH54]Sheet1!WT | | [1*:5*] | |
| Q46A | | | [DatapH54]Sheet1!Q46A | | [1*:7*] | |
| K188A | | | [DatapH54]Sheet1!K188A | | [1*:6*] | |
| K188E | | | [DatapH54]Sheet1!K188E | | [1*:8*] | |
| R192E | | | [DatapH54]Sheet1!R192E | | [1*:10*] | |
| Q46AK188A | | | [DatapH54]Sheet1!Q46AK188A | | [1*:7*] | |
| K188AR192A | | | [DatapH54]Sheet1!K188AR192A | | [1*:7*] | |
| K188AR259Q | | | [DatapH54]Sheet1!K188AR259Q | | [1*:7*] | |

| Variant | N | Min | Q1 | Median | Q3 | Max |
|---|---|---|---|---|---|---|
| WT | 5 | −7.7 | −6.7 | −5.7 | −3.2 | −2.7 |
| Q46A | 7 | −8.7 | −6.7 | −5.7 | −3.7 | −2.7 |
| K188A | 6 | −8.7 | −6.45 | −4.7 | −3.95 | −1.7 |
| K188E | 7 | −9.7 | −4.7 | −3.7 | −2.7 | −2.7 |
| R192E | 8 | −9.7 | −6.45 | −4.2 | −2.2 | −0.7 |
| Q46AK188A | 7 | −8.7 | −6.7 | −4.7 | −3.7 | −2.7 |
| K188AR192A | 7 | −10.7 | −7.7 | −4.7 | −2.7 | −1.7 |
| K188AR259Q | 7 | −11.7 | −4.7 | −1.7 | 0.3 | 2.3 |

| Variant | N | | Mean Rank | | Sum Rank | |
|---|---|---|---|---|---|---|
| WT | 5 | | 24.4 | | 122 | |
| Q46A | 7 | | 21.71429 | | 152 | |
| K188A | 6 | | 24.75 | | 148.5 | |
| K188E | 7 | | 30.5 | | 213.5 | |
| R192E | 8 | | 28.875 | | 231 | |
| Q46AK188A | 7 | | 24.85714 | | 174 | |
| K188AR192A | 7 | | 25.21429 | | 176.5 | |
| K188AR259Q | 7 | | 38.21429 | | 267.5 | |
| Chi-Square | DF | | Prob > Chi-Square | | | |
| 5.33505 | 7 | | 0.61915 | | | |

Null Hypothesis: The samples come from the same population.

Alternative Hypothesis: The samples come from different populations.

At the 0.05 level, the populations are NOT significantly different.

DOI: https://doi.org/10.7554/eLife.41741.025

## The ENS triad

E68, a glutamyl residue near the Schiff base constriction in the channel, forms an H-bond network with N239 and S43 (*Figure 5C*) with a geometry similar to that of a homologous triad (E129/E90, N297/N258, and S102/S63) referred to as 'the central gate' in C1C2 and *Cr*ChR2. In the CCRs, the triad blocks the cation path from the extracellular bulk phase (*Deisseroth and Hegemann, 2017*) and the glutamyl residue contributes to cation selectivity over anions (*Wietek et al., 2014*). In contrast, in *Gt*ACR1 the ENS triad does not occlude the tunnel (*Figure 5C*), but E68 is functionally important in channel gating and may serve as a Schiff base proton acceptor at least at basic pH (*Sineshchekov et al., 2015*). The three residues in the ENS triad appear to have distinct roles; that is the substitution S43A had little effect on Cl⁻ conductance, whereas the mutation N239A nearly eliminated the photocurrent (*Figure 5D*). Both S43A and N239A mutations decreased the rate of the slow channel closing, but N239A in addition strongly accelerated the fast decay (*Figure 5—figure*

**Table 4.** The output of Kriskal-Wallis ANOVA analysis of the results shown in *Figure 3—figure supplement 9B*

| X-Function | | | Kruskal-Wallis ANOVA | | |
|---|---|---|---|---|---|
| Data filter | | | No | | |
| Variant | | Data | | Range (number of cells) | |
| WT | | [DataNa1PipNa]Sheet1!WT | | [1*:11*] | |
| Q46A | | [DataNa1PipNa]Sheet1!Q46A | | [1*:6*] | |
| K188A | | [DataNa1PipNa]Sheet1!K188A | | [1*:10*] | |
| K188E | | [DataNa1PipNa]Sheet1!K188E | | [1*:6*] | |
| R192E | | [DataNa1PipNa]Sheet1!R192E | | [1*:8*] | |
| Q46AK188A | | [DataNa1PipNa]Sheet1!Q46AK188A | | [1*:7*] | |
| K188AR192A | | [DataNa1PipNa]Sheet1!K188AR192A | | [1*:8*] | |
| K188AR259Q | | [DataNa1PipNa]Sheet1!K188AR259Q | | [1*:10*] | |

| Variant | N | Min | Q1 | Median | Q3 | Max |
|---|---|---|---|---|---|---|
| WT | 11 | −11 | -9 | -5 | -3 | 0 |
| Q46A | 6 | −11 | −8.75 | −2.5 | −0.5 | 1 |
| K188A | 10 | -9 | −5.5 | -4 | -4 | -2 |
| K188E | 6 | −14 | −12.5 | −6.5 | −3.75 | -3 |
| R192E | 8 | -7 | −6.75 | -5 | −2.5 | -2 |
| Q46AK188A | 7 | -9 | -9 | -5 | -3 | 1 |
| K188AR192A | 8 | −16 | −13.25 | -4 | −0.5 | 7 |
| K188AR259Q | 10 | −15 | −9.25 | −7.5 | -5 | -3 |

| Variant | N | | Mean Rank | | Sum Rank |
|---|---|---|---|---|---|
| WT | 11 | | 34.13636 | | 375.5 |
| Q46A | 6 | | 43.33333 | | 260 |
| K188A | 10 | | 37 | | 370 |
| K188E | 6 | | 26.33333 | | 158 |
| R192E | 8 | | 36.75 | | 294 |
| Q46AK188A | 7 | | 35.5 | | 248.5 |
| K188AR192A | 8 | | 35.3125 | | 282.5 |
| K188AR259Q | 10 | | 22.25 | | 222.5 |
| Chi-Square | DF | | Prob > Chi-Square | | |
| 6.6454 | 7 | | 0.46671 | | |

Null Hypothesis: The samples come from the same population.

Alternative Hypothesis: The samples come from different populations.

At the 0.05 level, the populations are NOT significantly different.

DOI: https://doi.org/10.7554/eLife.41741.026

*supplement 3A*). Remarkably, combining the N239A mutation with the D234N mutation which alone also accelerated the fast decay, returned the channel closing kinetics almost to that of the wild type (*Figure 5—figure supplement 3B*). Given its location between C2 and C3, N239 may assist moving anions between the Schiff base and the cytoplasmic port (*Figure 2C*). Additionally, the distribution of apolar residues in this portion of the channel would also facilitate quick movements of anions as has been proposed for the CLC channel (*Park and MacKinnon, 2018*).

Despite the large phylogenetic difference between cryptophyte ACRs and chlorophyte CCRs, their helical scaffolds are little changed. However, the *Gt*ACR1 structure reveals fundamentally different chemistry built within their common scaffold. The preexisting full-length tunnel, the location of the retinylidene photoactive site directly in the ion path, the maintenance of a net positive charge on the site's Schiff base in a largely neutral tunnel, and the novel extracellular cap, provide important clues to the structural basis of light-gated anion conductance.

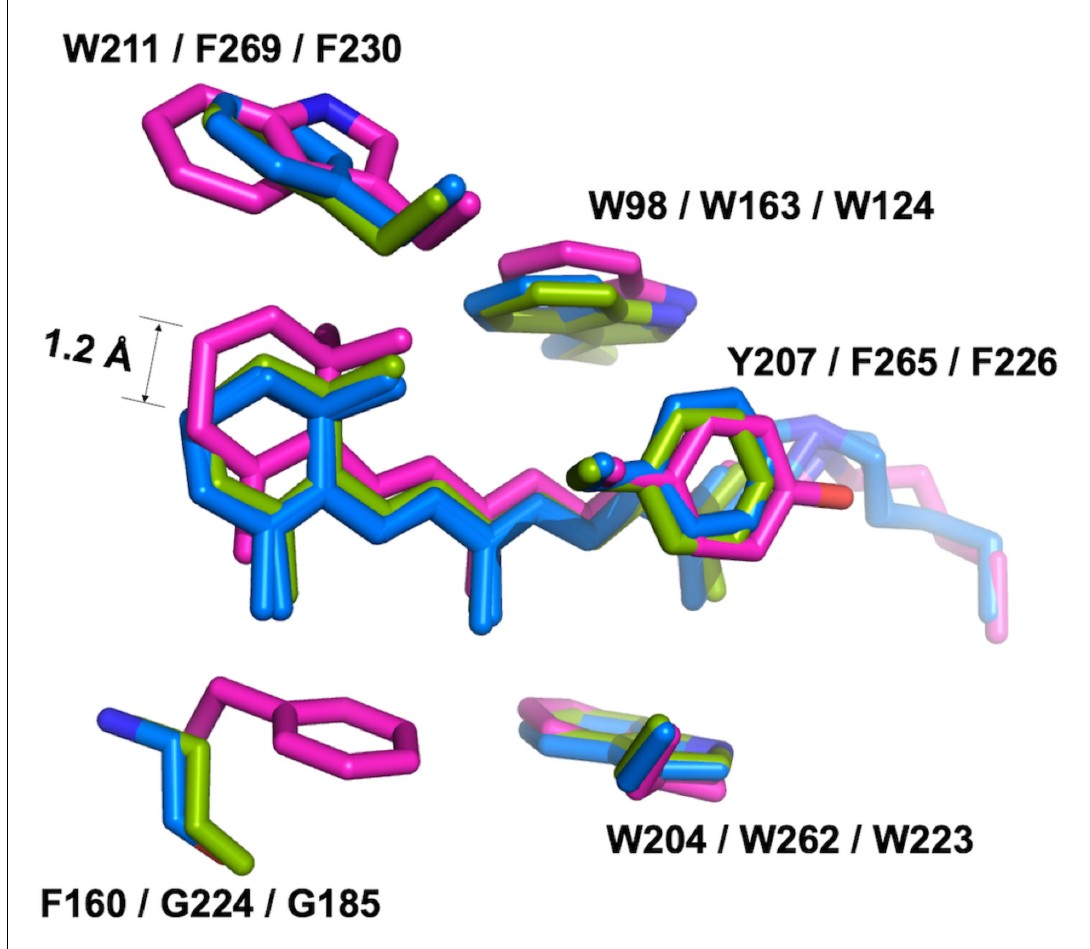

**Figure 4.** The retinal conformation. The structure of *Gt*ACR1 (*magenta*) is superimposed with C1C2 (*green*) and *Cr*ChR2 (*blue*) using SSM. The presence of F160 in *Gt*ACR1 (G224/G185 in C1C2/*Cr*ChR2, respectively) pushes the β-ionone ring of the all-*trans* retinal towards the extracellular side by 1.2 Å as measured between the C16 atoms of *Gt*ACR1 and *Cr*ChR2.

DOI: https://doi.org/10.7554/eLife.41741.027

The following figure supplement is available for figure 4:

**Figure supplement 1.** The action spectra of photocurrents generated by the F160G mutant as compared with wild-type *Gt*ACR1.

DOI: https://doi.org/10.7554/eLife.41741.028

## Comparison with the recently published *Gt*ACR1 structure

We report the atomic structure of the 7-helix rhodopsin domain (residues 1–295) that contains the light-gated channel activity of *Gt*ACR1. After this manuscript was prepared for submission, an article reporting a structure of the same domain with a short truncation (PDB code: 6CSM, residues 1–282) appeared from Karl Deisseroth and coworkers (*Kim et al., 2018*).

The two *Gt*ACR1 structures were both determined at 2.9 Å resolution using the lipid cubic phase crystallization method. However, they were obtained in different crystallization conditions and exhibit different space groups and crystal packing. Nevertheless, the two show an extremely high similarity with a marginal RMSD value of ~0.4 Å. All seven transmembrane helices are very well superimposed between the two structures (*Figure 5—figure supplement 4*). The truncation or different crystal packing did not give rise to any large differences between the two protein structures. Therefore, these two structures presented by two independent groups mutually validate the conformation of *Gt*ACR1 in the closed state.

Differences between our study and that of *Kim et al., 2018* are primarily in the methods used to deduce the location of the anion conduction pathway in *Gt*ACR1 and the results of testing relative ionic permeabilities of the mutants. By examining the structure by serial cross-sectioning we

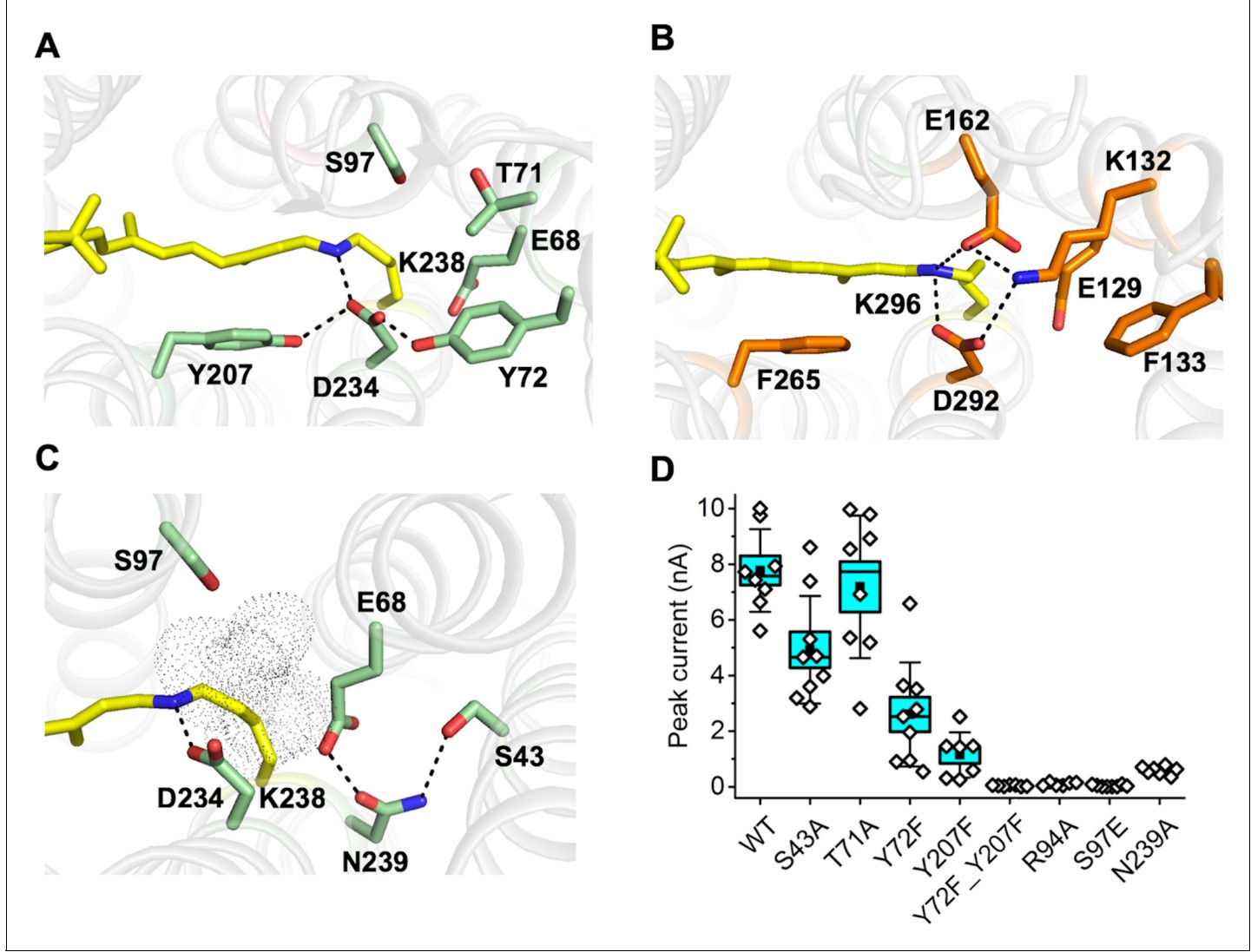

**Figure 5.** Conformation of the Schiff base region of *Gt*ACR1. (A–B) Structural comparison shows different H-bond networks (*dashed* lines) in *Gt*ACR1 (A) and C1C2 (B). (C) the H-bond network in the ENS triad of *Gt*ACR1. The tunnel (*black* dots) assessed by CAVER. (D) Peak photocurrents generated by the wild-type *Gt*ACR1 and indicated mutants in response to laser flash excitation. The black squares, mean; line, median; box, SE; whiskers, SD; empty diamonds, raw data recorded from individual cells.

DOI: https://doi.org/10.7554/eLife.41741.029

The following source data and figure supplements are available for figure 5:

**Source data 1.** Numerical data for the current amplitude values measured in individual cells are shown in *Figure 5D*.
DOI: https://doi.org/10.7554/eLife.41741.034

**Figure supplement 1.** The action spectra of photocurrents generated by the Y72F and Y207F mutants, as compared to the wild type.
DOI: https://doi.org/10.7554/eLife.41741.030

**Figure supplement 2.** Laser flash-evoked photocurrents generated by the Y72F and Y207F mutants, as compared to the wild type.
DOI: https://doi.org/10.7554/eLife.41741.031

**Figure supplement 3.** Laser flash-evoked photocurrents generated by.
DOI: https://doi.org/10.7554/eLife.41741.032

**Figure supplement 4.** Superposition of two independently obtained *Gt*ACR1 structures.
DOI: https://doi.org/10.7554/eLife.41741.033

identified a contiguous intramolecular tunnel from the extracellular to intracellular surfaces. The CAVER algorithm confirmed the tunnel and helped identify three constrictions. Beyond the electropositive extracellular port the tunnel is lined primarily by small polar and aliphatic residues with sparse positive regions, one of them the protonated retinylidene Schiff base itself. Kim et al. described the putative ion conduction pathway in *Gt*ACR1 by identifying intramolecular cavities (vestibules) with electropositive surfaces, which led to identification of mostly residues present in the tunnel walls. Deeper understanding of the foundations of anion conductance by ACRs will likely require an atomic structure of the open state conformation.

Regarding anion selectivity of the mutants, a difference between our study and that of Kim et al.'s is that we have found no detectable cation permeability in any so far examined mutants (*Figure 3—figure supplements 3–9*). We attribute the strict selectivity for anions, a notable property of ACRs, to the existence of multiple highly selective portions of the tunnel and its two entry/exit ports. Consistent with this view, Kim et al. also concluded that there are multiple selective regions rather than a single selectivity filter, in contrast to CCRs, in which cation selectivity can be weakened to allow partial anion permeability by mutation of even a single residue (*Wietek et al., 2014*).

Overall, the independent analyses of the structures by both groups provide complimentary information creating a firm basis for elucidating the functional mechanism and for further optimization of *Gt*ACR1 as a potent neuron-silencing optogenetic tool.

# Materials and methods

**Key resources table**

| Reagent type (species) or resource | Designation | Source or reference | Identifiers | Additional information |
|---|---|---|---|---|
| Gene (*Guillardia theta*) | *Gt*ACR1 | Synthetic | GenBank: KP171708 | |
| Cell line (*Spodoptera frugiperda*) | Sf9 | Sigma Aldrich | Sigma Aldrich: 89070101, RRID:CVCL_0549 | |
| Cell line (*Homo sapiens*) | HEK293 | ATCC | ATCC: CRL-1573, RRID:CVCL_0045 | |
| Recombinant DNA reagent | Cellfectin II Reagent | Thermo Fisher | Cat. No.: 10362100 | https://www.thermofisher.com/order/catalog/product/10362100 |
| Recombinant DNA reagent | ScreenFectA transfection reagent | Waco Chemicals USA | Cat. No.: 299–73203 | http://www.e-reagent.com/uh/Shs.do?now=1544459665328 |
| Recombinant DNA reagent | pFastbac1 | Thermo Fisher | Cat. No.: 10360014 | https://www.thermofisher.com/order/catalog/product/10360014 |
| Recombinant DNA reagent | pcDNA3.1 | Thermo Fisher | Cat. No.: V79020 | https://www.thermofisher.com/order/catalog/product/V79020 |
| Software, algorithm | Pymol | PyMOL Molecular Graphics System, Schrödinger, LLC | RRID:SCR_000305 | http://www.pymol.org/ |
| Software, algorithm | UCSF Chimera | UCSF Resource for Biocomputing, Visualization, and Bioinformatics | RRID: SCR_004097 | http://plato.cgl.ucsf.edu/chimera/ |

*Continued on next page*

*Continued*

| Reagent type (species) or resource | Designation | Source or reference | Identifiers | Additional information |
|---|---|---|---|---|
| Software, algorithm | PHENIX | PMID: 20124702 | RRID: SCR_014224 | http://www.phenix-online.org/ |
| Software, algorithm | Coot | PMID: 15572765 | RRID: SCR_014222 | http://www.biop.ox.ac.uk/coot/ |
| Software, algorithm | OriginPro 2016 | OriginLab | | https://originlab.com |
| Software, algorithm | pClamp 10 | Molecular Devices | RRID:SCR_011323 | http://www.moleculardevices.com/products/software/pclamp.html |

## GtACR1 expression from insect cells and purification

*Gt*ACR1 protein was expressed and purified from *Spodoptera frugiperda* Sf9 cells using a baculovirus expression system. The *Gt*ACR1 gene (GenBank Acc. KP171708, amino acid sequence 1–295) was fused with a C-terminal His8 tag and subcloned into the plasmid pFastbac1 (Invitrogen) between the cloning sites *EcoR*I and *Xba*I. Baculovirus were generated by following a standard protocol (Thermofisher, Waltham, MA). To express the *Gt*ACR1 protein, exponentially-grown S9 cells (cell density $\sim 2 \times 10^6$/ml) were infected by *Gt*ACR1-encoding virus in a ratio of 15:1 (v/v). All-*trans*-retinal in ethanol was added to the culture to the final concentration 5 µM. Cells were incubated for 3 days in spinner flasks at 27°C. The pink-colored cells were harvested by centrifugation using an SS34 rotor at 3000 rpm for 10 min, and the cell pellets were resuspended in Buffer A (350 mM NaCl, 5% glycerol, 20 mM HEPES, pH 7.5) with 0.1 mM phenylmethylsulfonyl fluoride (PMSF). Cell rupture was performed by 3 × passage through a high-pressure homogenizer EmulsiFlex-C3 (Avestin, Ottawa, ON). After centrifugation at low speed (5000 rpm for 10 min) to remove cell debris, membrane fractions were pelleted at 40,000 rpm for 1 hr using a Ti45 Beckman rotor. The membranes were suspended in Buffer A and solubilized with 1% dodecyl-maltoside (DDM) for 1 hr at 4°C with shaking. Undissolved content was removed after ultracentrifugation using a Ti45 rotor at 45,000 rpm for 1 hr. The supernatant supplemented with 15 mM imidazole was incubated with Ni resin (Qiagen, Hilden, Germany) for 1 hr with shaking at 4°C. The resin was step-wise washed using 15 mM and 40 mM imidazole in Buffer A supplemented with 0.03% DDM. The *Gt*ACR1 protein was eluted with 400 mM imidazole and 0.03% DDM in buffer A. The eluted protein was further purified using a Superdex Increase 10/300 GL column (GE Healthcare, Chicago, IL) equilibrated with Buffer B (350 mM NaCl, 5% glycerol, 0.03% DDM, 20 mM MES, pH 5.5). Protein fractions with an A280/A515 absorbance ratio of ~1.9 were pooled, concentrated to ~20 mg/ml using a 100 K MWCO filter, flash-frozen in liquid nitrogen and stored at −80°C until use. Molar protein concentration was calculated using the absorbance value at 515 nm divided by the extinction coefficient 45,000 $M^{-1}cm^{-1}$.

## Protein crystallization

Crystallization was carried out using the in meso approach. 40 µl of *Gt*ACR1 protein was mixed with 60 µl monoolein (MO) (Sigma, St. Louis, MO or Nu-chek, Waterville, MN), premelted at 42°C using two syringes until the mixture became transparent to form a lipidic mesophase (lipidic cubic phase; LCP). 150 nl aliquots of the protein-mesophase mixture were spotted on 96-well LCP glass sandwich plates (Molecular Dimensions, Maumee OH) and overlaid with 1.5 µl of precipitant solution using a Gryphon crystallization robot (Art Robbins, Sunnyvale, CA). The plates were covered by aluminum foil to maintain them dark and incubated at room temperature. Red-colored *Gt*ACR1 crystals of ~20 µm in size appeared after one month. The most highly diffracting crystals were obtained in a protein-mesophase mixture containing 15% 2-methyl-2,4-pentanediol (MPD), 0.1 M 2-[(2-amino-2-oxoethyl)-(carboxymethyl)amino]acetic acid (ADA), pH 6.0, and 0.1 M NaCl. Crystals in LCP were harvested using micromesh loops (MiTeGen, Ithaca, NY), and flash-cooled in liquid nitrogen without any additional cryoprotectant.

## Data collection and processing

X-ray diffraction data collections were performed on protein crystallography beamlines X06SA-PXI at the Swiss Light Source (SLS), Villigen, Switzerland. Data were collected with a $10 \times 10\ \mu m^2$ microfocused X-ray beam of 12.398 keV (1 Å in wavelength) at 100 K using SLS data acquisition software suites (DA+) (*Wojdyla et al., 2018*). Continuous grid-scans were used to locate crystals in frozen LCP samples (*Wojdyla et al., 2016*). The collection strategy was in steps of 0.1° at a speed of 0.1 s/step by using the EIGER 16M detector operated in continuous/shutterless data collection mode. Data were processed with XDS and scaled and merged with XSCALE (*Kabsch, 2010a*; *Kabsch, 2010b*). Four partial data sets (three with 60° wedges and one with 30° wedges) were collected, processed and merged to a final data set to 2.9 Å resolution. Data collection and processing statistics are provided in *Table 1*.

## Structure determination

The *Gt*ACR1 structure was determined using the molecular replacement (MR) method with the structure of *Chlamydomonas reinhardtii* ChR2 (PDB entry: 6EID) (*Volkov et al., 2017*) as the search model. The MR solution was obtained using Phaser (*McCoy et al., 2007*) with the TFZ to 8.7 and LLG to 221. The initial model was built using PHENIX-autobuild (*Adams et al., 2010*) and further completed manually using COOT (*Emsley and Cowtan, 2004*). The structural refinement was performed using PHENIX (*Adams et al., 2010*) The final structure has $R_{work}/R_{free}$ factors of 0.25/0.27. Refinement statistics are reported in *Table 1*. The structure factors and coordinates have been deposited in the Protein Data Bank (PDB entry code: 6EDQ). Figures of molecular structures were generated with PyMOL (http://www.pymol.org).

## *Gt*ACR1 expression and electrophysiology

Characterization of *Gt*ACR1 mutants was performed using whole-cell photocurrent recording as previously described (*Sineshchekov et al., 2015*). Briefly, the wild-type expression construct was cloned into the mammalian expression vector pcDNA3.1 (Life Technologies, Carlsbad, CA) in frame with an EYFP (enhanced yellow fluorescent protein). Mutations were introduced using a QuikChange XL site-directed mutagenesis kit (Agilent Technologies, Santa Clara, CA) and verified by DNA sequencing. HEK293 (human embryonic kidney) cells were transfected using the ScreenFectA transfection reagent (Waco Chemicals USA, Richmond, VA). All-*trans*-retinal (Sigma, St. Louis, MO) was added at the final concentration 4 μM immediately after transfection. Photocurrents were recorded 48–72 hr after transfection in whole-cell voltage clamp mode at room temperature (25°C) with an Axopatch 200B amplifier (Molecular Devices, Union City, CA) and digitized with a Digidata 1440A using pClamp 10 software (both from Molecular Devices). Currents recorded in response to laser excitation or continuous light were filtered with a 10 or 2 kHz low-pass Bessel filter and digitized at 250 or 5 kHz, respectively. Patch pipettes with resistances of 2–5 MΩ were fabricated from borosilicate glass and filled with the following solution (in mM): KCl 126, $MgCl_2$ 2, $CaCl_2$ 0.5, EGTA 5, HEPES 25, and pH 7.4. The standard bath solution contained (in mM): NaCl 150, $CaCl_2$ 1.8, $MgCl_2$ 1, glucose 5, HEPES 10, pH 7.4. To test for changes in the permeability for $Cl^-$, this ion in the bath was partially replaced with non-permeable aspartate (the final $Cl^-$ concentration 5.6 mM, rounded to 6 mM in the figure legends). To test for changes in the permeability for $H^+$, the bath pH was adjusted to 5.4, and for $Na^+$, this ion in the bath was partially replaced with N-methyl-D-gluconate (NMDG) neutralized with $H_2SO_4$. (the final $Na^+$ concentration 1.4 mM). In this latter case, $K^+$ in the pipette was fully replaced with $Na^+$, so that $Na^+$ was the only monovalent metal cation present in the system. A 4 M KCl bridge was used in all measurements. Series resistance was periodically checked during recording, and measurements showing >20% increase were discarded. The current-voltage relationships (IE curves) were measured near the expected $E_{rev}$ to eliminate its possible changes during recording. For each cell, one value of the $E_{rev}$ was calculated. Before averaging, the curves for individual cells were normalized to the value obtained at the most negative holding potential in the tested range. The holding potential values were corrected for liquid junction potentials calculated using the Clampex built-in LJP calculator (*Barry, 1994*). Laser excitation was provided by a Minilite Nd:YAG laser (532 nm, pulsewidth six ns, energy 12 mJ; Continuum, San Jose, CA). A laser artifact measured with a blocked optical path was digitally subtracted from the recorded traces. For further analysis, the signals were logarithmically averaged with a custom-created computer algorithm. Curve fitting and

data analysis were performed using OriginPro 2016 software (OriginLab Corporation, Northampton, MA). Continuous light pulses were provided by a Polychrome V light source (T.I.L.L. Photonics GMBH, Grafelfing, Germany) at 15 nm half-bandwidth in combination with a mechanical shutter (Uniblitz Model LS6, Vincent Associates, Rochester, NY; half-opening time 0.5 ms). The maximal quantum density at the focal plane of the $40 \times$ objective measured with a piezo detector was 7.7 mW mm$^{-2}$ at 515 nm. For measurements of the action spectra, short (25 ms) light pulses of the intensity in the linear response range were used at 10 nm half-bandwidth. The mean currents during the initial rise were calculated and corrected for the quantum density at each wavelength, which was measured with a calibrated photodiode. In each cell, a response to illumination at each wavelength was measured at least twice in a symmetrical fashion, first, scanning from the shortest to the longest wavelength, and then in the opposite direction. The spectral data sets obtained in all scans were pooled together (because the differences between individual cells in ACR expression levels or patch parameters were not expected to influence ACR spectral properties), normalized to the maximal value and averaged to produce the mean and sem values.

Transfection with each tested mutant variant was repeated in at least three different batches of culture, and the results obtained in cells from all batches were pooled. Batches of culture were randomly allocated for transfection with a specific mutant; no masking (blinding) was used. Individual transfected HEK293 cells were selected for patching by inspecting their tag fluorescence; non-fluorescent cells were excluded. Cells for which we could not establish a gigaohm seal or for which a gigaohm seal was lost during recording were excluded from measurements. Current traces recorded from the same cells upon repetitive light stimulation were considered as technical replicates; results obtained from different individual cells were considered as biological replicates. In experiments with laser excitation, 10 technical replicates were averaged to yield a single mean trace for each cell; in experiments with continuous light pulses, a single trace was recorded in each cell. The baseline measured before illumination was subtracted using Clampfit software (a subroutine of pClamp). The same software was used to measure the peak current amplitude with a cursor. The raw data obtained in individual cells are shown as open diamonds and listed in the corresponding source data tables. Sample size was estimated from previous experience and published work on a similar subject, as recommended by the NIH guidelines (*Dell et al., 2002*). No outliers were excluded from calculation of mean values. Normality of the data was not assumed, and therefore non-parametric statistical tests were used as implemented in OriginPro 2016 software; P values > 0.05 were considered not significant. The results of statistical hypothesis testing are shown in *Tables 2–4*. When no specific statistical hypothesis was tested, descriptive statistical analysis was applied.

## Cell lines

Only commercially available cell lines authenticated by the vendors (Sf9 from Sigma Aldrich and HEK293 from ATCC) were used; no cell lines from the list of commonly misidentified cell lines were used. The absence of micoplasma contamination was verified by Visual-PCR mycoplasma detection kit (GM Biosciences, Frederick, MD).

## Data availability

Atomic coordinates and structure factors for the reported crystal structure have been deposited with the Protein Data Bank (PDB) under the accession code 6EDQ.

## Acknowledgements

This work was supported by National Institutes of Health Grants R01GM027750 and U01MH109146, the Hermann Eye Fund, and Endowed Chair AU-0009 from the Robert A Welch Foundation to JLS, and American Heart Association Grant 18TPA34230046 to LZ. C-YH was partially supported by the European Union's Horizon 2020 research and innovation programme under the Marie-Skłodowska-Curie grant agreement No. 701647.

## Additional information

### Competing interests

Elena G Govorunova: EGG and The University of Texas Health Science Center at Houston have filed patent applications that relate to ACRs (PCT application PCT/US2016/023095, entitled Compositions And Methods For Use Of Anion Channel Rhodopsins). Oleg A Sineshchekov: OAS and The University of Texas Health Science Center at Houston have filed patent applications that relate to ACRs (PCT application PCT/US2016/023095, entitled Compositions And Methods For Use Of Anion Channel Rhodopsins). John L Spudich: JLS and The University of Texas Health Science Center at Houston have filed patent applications that relate to ACRs (PCT application PCT/US2016/023095, entitled Compositions And Methods For Use Of Anion Channel Rhodopsins). The other authors declare that no competing interests exist.

### Funding

| Funder | Grant reference number | Author |
| --- | --- | --- |
| National Institutes of Health | R01GM027750 | John L Spudich |
| National Institutes of Health | U01MH109146 | John L Spudich |
| Welch Foundation | AU-0009 | John L Spudich |
| American Heart Association | 18TPA34230046 | Lei Zheng |
| Hermann Eye Fund | | John L Spudich |
| H2020 Marie Skłodowska-Curie Actions | 701647 | Chia-Ying Huang |

The funders had no role in study design, data collection and interpretation, or the decision to submit the work for publication.

### Author contributions

Hai Li, Data curation, Software, Formal analysis, Validation, Investigation, Visualization, Methodology, Writing—original draft; Chia-Ying Huang, Data curation, Software, Formal analysis, Funding acquisition, Validation, Investigation, Visualization, Methodology, Writing—original draft; Elena G Govorunova, Data curation, Software, Formal analysis, Validation, Investigation, Visualization, Methodology, Writing—original draft, Writing—review and editing; Christopher T Schafer, Software, Formal analysis, Validation, Visualization, Methodology, Writing—original draft; Oleg A Sineshchekov, Software, Formal analysis, Validation, Investigation, Visualization, Writing—original draft; Meitian Wang, Conceptualization, Resources, Data curation, Software, Formal analysis, Supervision, Funding acquisition, Validation, Investigation, Visualization, Methodology, Writing—original draft; Lei Zheng, John L Spudich, Conceptualization, Resources, Software, Formal analysis, Supervision, Funding acquisition, Validation, Investigation, Visualization, Methodology, Writing—original draft, Project administration, Writing—review and editing

### Author ORCIDs

Hai Li ID http://orcid.org/0000-0002-3969-6709
Chia-Ying Huang ID http://orcid.org/0000-0002-7676-0239
Lei Zheng ID https://orcid.org/0000-0001-7789-5234
John L Spudich ID http://orcid.org/0000-0003-4167-8590

### Decision letter and Author response

Decision letter https://doi.org/10.7554/eLife.41741.039
Author response https://doi.org/10.7554/eLife.41741.040

## Additional files

### Supplementary files

• Transparent reporting form
DOI: https://doi.org/10.7554/eLife.41741.035

### Data availability

Diffraction data have been deposited in PDB under the accession code 6EDQ.

The following dataset was generated:

| Author(s) | Year | Dataset title | Dataset URL | Database and Identifier |
|---|---|---|---|---|
| Li H, Huang CY | 2018 | Crystal Structure of the Light-Gated Anion Channelrhodopsin GtACR1 | http://www.rcsb.org/structure/6EDQ | Protein Data Bank, 6EDQ |

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
