## [Decision Letter]

Thank you for submitting your article "Crystal structure of a natural light-gated anion channelrhodopsin" for consideration by *eLife*. Your article has been reviewed by Richard Aldrich as the Senior Editor, a Reviewing Editor, and three reviewers. The reviewers have opted to remain anonymous.

The reviewers have discussed the reviews with one another and the Reviewing Editor Dr. Sriram Subramaniam has drafted this decision to help you prepare a revised submission.

All of the reviewers agreed that this is a very well executed manuscript with convincing experimental evidence. The reviewers also recognized that your manuscript reports the crystal structure of the full-length protein, independently verifying the previously obtained structure of a truncated form of this channel, thus creating a firm basis for further optimization of a potent neuron-silencing optogenetic tool.

Essential revisions:

1) Effects of crystal packing on the structure that may arise from different crystallization constructs.

2) The conclusion that (i) the retinylidene Schiff base protonated configuration is maintained when the channel is open and (ii) Schiff base controls gating and also serves as a direct mediator for anion flux is not directly demonstrated. It would be good to address this if possible.

3) Kinetic analysis of channel open/close events is not presented. As a result, one cannot distinguish between different loss-of-function mutant phenotypes affecting C1, C2 and C3 regions.

4) Spectral characteristics of mutants are lacking.

5) Figure 1—figure supplement 1 should include spectral and functional (anion transport) validation data for natural vs recombinant purified ACR1 proteins.

6) Subsection “Ion pathway constrictions”: “dark-grown crystals is presumably the dark (closed) state of the channel protein”, is this correct?

7) Subsection “Ion pathway constrictions”: What is the rationale for choice of Glu for site-specific replacements? Why was there no consideration of gain-of-function testing? Could the use of Glu simply change electrochemical repulsion of anion flux rather than physically opening the channel?

8) Did the M105E mutant cause photochemical perturbation as this residue is closed to Schiff base linkage?

9) If possible, please show conductance, current/voltage relationships and reversal potentials for the crystalized construct, native protein and individual mutants as supplementary data.

a) Absorption spectra and kinetics of wild-type GtACR1 and mutants should be shown in the supplement.

b) Especially for the Schiff base protonation in wild-type and mutants, it would be good to present detailed spectroscopic data.

c) The authors should include a discussion paragraph highlighting key unique features that distinguish their work from the work of Kim et al., (2018).

---

## [Author Response]

All of the reviewers agreed that this is a very well executed manuscript with convincing experimental evidence. The reviewers also recognized that your manuscript reports the crystal structure of the full-length protein, independently verifying the previously obtained structure of a truncated form of this channel, thus creating a firm basis for further optimization of a potent neuron-silencing optogenetic tool.Essential revisions:1) Effects of crystal packing on the structure that may arise from different crystallization constructs.

We have added a subsection “Comparison with the recently published *Gt*ACR1 structure” at the end of Results and Discussion section. The two structures show an extremely high similarity with a marginal RMSD value of ~0.4 Å. All seven transmembrane helices are very well superimposed between the two structures. We added a figure showing their superimposition (Figure 5—figure supplement 4). Neither the different constructs nor different crystal packing gave rise to any large differences in the protein structure.

2) The conclusion that (i) the retinylidene Schiff base protonated configuration is maintained when the channel is open and (ii) Schiff base controls gating and also serves as a direct mediator for anion flux is not directly demonstrated. It would be good to address this if possible.

We suggest the interesting possibility of direct participation of the protonated Schiff base based on several observations and have expanded the previous paragraph in the text, which now reads as follows:

“The location of the protonated Schiff base (PSB), centered within the anion path in *Gt*ACR1, suggests that it may play a direct role in anion translocation in the open channel state. Consistent with this idea is that the PSB is only partially neutralized by its D234 counterion because the counterion is weakened by its interaction with the 2 tyrosinyl residues. Therefore, the PSB presents a partial positive charge capable of transient reversible interaction with Cl^-^ ions in a channel that is largely lined by small polar and hydrophobic aliphatic residues. Supporting a possible direct role of the PSB, late deprotonation of the Schiff base after channel opening occurs in all three ACRs so far examined: *Gt*ACR1 and *Gt*ACR2 (Sineshchekov et al., 2016) and *Psu*ACR1 (Wietek et al., 2016), yet Schiff base deprotonation after channel opening is not known to occur in any CCR. Further indicating an essential role of the protonated form, the mutant S97E, in which a potential Schiff base proton acceptor is placed at the corresponding position in *Gt*ACR1 as in CCRs and many other microbial rhodopsins, exhibits (i) appearance of fast Schiff base deprotonation, and (ii) a >30-fold suppression of the amplitude of the chloride photocurrent (Sineshchekov et al., 2016). Furthermore, the double mutation Y207F/Y72F, expected to release inhibition of D234 as a proton acceptor, decreased the photocurrent amplitude to a negligible value (Figure 5D).”

3) Kinetic analysis of channel open/close events is not presented. As a result, one cannot distinguish between different loss-of function mutant phenotypes affecting C1, C2 and C3 regions.

We have added a new figure (Figure 3—figure supplement 2) to the revision showing photocurrent traces from the mutants in the C1 and C3 regions recorded under single-turnover conditions (i.e. upon laser flash excitation), and the results of their multiexponential fitting and the time constants of the individual kinetic components. Kinetics of the photocurrents from the C2 mutants could not be analyzed because of their greatly suppressed amplitude (Figure 3G).

4) Spectral characteristics of mutants are lacking.

Spectral characteristics of the wild-type and mutants in key sites are presented in Sineshchekov et al., 2016 referenced in this manuscript. We have added with discussion in the revision two new figures (Figure 4—figure supplement 1 and Figure 5—figure supplement 1) showing the action spectra of photocurrents generated by the F150G, Y72F and Y207F mutants.

5) Figure 1—figure supplement 1 should include spectral and functional (anion transport) validation data for natural Vs recombinant purified ACR1 proteins.

Unlike bacteriorhodopsin and several other type 1 rhodopsins, channelrhodopsins are present in algal cells at very low concentrations (e.g. Govorunova et al., (2004)), so that not only purification of natural channelrhodopsins from algal cells, but also their spectroscopic detection in situ are technically not feasible. Therefore, only recombinant *Gt*ACR1 expressed in heterologous systems have so far been characterized (Sineshchekov et al., 2015; 2016 referred to in this manuscript and other articles referred to in Govorunova et al., 2017).

The wild-type expression construct (encoding residues 1-295) used for crystallization in this study was the same as used in our previous electrophysiological recordings in cultured animal cells and as used for spectroscopic and photochemical characterization of the protein expressed and purified from *Pichia* (Sineshchekov et al., 2015; 2016; Govorunova et al., 2017, and references therein). The wild-type photocurrent action spectrum in HEK293 cells is superimposable with the absorption spectrum from *Pichia*-expressed purified *Gt*ACR1 (Sineshchekov et al., 2015, 2016) which is identical to the absorption spectrum of purified insect (Sf9) cell-expressed GtACR1 used for crystallization.

For clarification in the revised manuscript we show the absorption spectrum of the purified crystallized construct in Figure 1—figure supplement 1C, and a representative photocurrent trace recorded upon laser flash excitation from this construct expressed in HEK293 is shown as a dashed line in Figure 3—figure supplements 1 and 2, Figure 4—figure supplement 1 and Figure 5—figure supplement 1.

6) Subsection “Ion pathway constrictions”: “dark-grown crystals is presumably the dark (closed) state of the channel protein”, is this correct?

We expect it to be closed because the crystals are formed in the dark and wild-type *Gt*ACR1 in membranes is in a tightly closed-channel conformation in the dark. The structure itself is consistent with the dark (closed) state. The intramolecular tunnel that traverses the protein is narrowly constricted at positions C1, C2, and C3 and further potentially blocked by the retinylidene Schiff base at C2 and by the extracellular cap.

7) Subsection “Ion pathway constrictions”: What is the rationale for choice of Glu for site-specific replacements? Why was there no consideration of gain-of-function testing? Could the use of Glu simply change electrochemical repulsion of anion flux rather than physically opening the channel?

Glu was initially as a substituent because two of the C1 residues correspond in position to highly conserved Glu residues in CCRs known to be important in cation conductance. Second, the negative charge and its bulky sidechain are expected to block the anion channel thus decreasing or eliminating photocurrents. We have not suggested that introduction of Glu would physically open the channel; rather we expected that it would block, and, as Figure 3G shows, this expectation turned out to be true for C2 and C3 constrictions, distinguishing them from C1 residues. For clarification we expanded the paragraph on the Glu scanning of constriction sites in the revision as follows:

“We chose Glu as a substituent because two of the constriction residues, A61 and A75, correspond to the highly conserved Glu residues in CCRs (E122/E83 and E136/E97 in C1C2/*Cr*ChR2, respectively), and because neutralization of E83 was required for elimination of the residual H^+^ permeability in Cl^-^-conducting CCR mutants (Berndt et al., 2016; Wietek et al., 2015). We also hypothesized that the bulky negatively charged Glu side chain would block the *Gt*ACR1 channel when placed in the ion conduction pathway. Indeed, perturbation of any residues at C2 or C3 greatly reduced or eliminated the photocurrents, while effects of most mutations (except A75E) at the C1 position were negligible (Figure 3G), suggesting that in the open conformation the channel is wider in the extracellular portion and more narrow in its central and intracellular stretches.”

It is not clear to us what “gain-of-function testing” is referred to in the question. Recording and analysis of photocurrents reveals both loss- and gain-of-function effects of mutations. In this regard, of all mutants we have tested in this study, none exhibited larger currents than the wild type (Figures 3G and Figure 5D).

8) Did the M105E mutant cause photochemical perturbation as this residue is closed to Schiff base linkage?

M105 is a constituent of the retinal binding pocket and is located adjacent to the Schiff base with a distance of 4.6 Å, so it is highly likely that its perturbation as well as that of other residues near the Schiff base would influence the photochemical reaction cycle. Study of photoactive site residues will be of interest, but we regard analysis of mutant photocycles as beyond the scope of this report.

9) If possible, please show conductance, current/voltage relationships and reversal potentials for the crystalized construct, native protein and individual mutants as supplementary data.

The crystallized construct and several of its mutants have been characterized in our previous publications (Sineshchekov et al., 2015 and 2016) referred to in this manuscript. As explained above, isolation of native *Gt*ACR1 protein from algal cells is not feasible. The conductance data (mean photocurrent amplitudes) of the mutants made in this study are shown in Figures 3G and 5D. In the revision we have added new figures (Figure 3—figure supplement 3 and Figure 3—figure supplement 4) showing the current-voltage relationships (IE curves) and reversal potentials for wild-type *Gt*ACR1 and mutants and discussion of these data in the text.

a) Absorption spectra and kinetics of wild-type GtACR1 and mutants should be shown in the supplement.

The absorption spectrum of the crystallized construct is shown in Figure 1—figure supplement 1C. A representative photocurrent trace recorded upon laser flash excitation from this construct expressed in HEK293 is shown as a dashed line in Figure 3 —figure supplements 1 and 2, Figure 4 —figure supplement 1 and Figure 5—figure supplement 1. Detailed analyses of wild-type current kinetics and photochemical reaction cycle transition kinetics have been reported in Sineshchekov et al., 2015 and 2016, respectively, referred to in this manuscript.

We have added to the revision four new figures (Figure 3—figure supplements 1 and 2, Figure 5—figure supplement 2 and Figure 5—figure supplement 3) showing the currents kinetics recorded under single-turnover conditions from the mutants described in this study, along with the results of their multiexponential fitting and numerical values of the derived time constants of individual current components. Two new figures (Figure—figure supplement 1 and Figure 5—figure supplement 1) have been added to the revision showing the action spectra of photocurrents generated by the mutants. Descriptions of these results have been added where these new figures are referred to in the text.

b) Especially for the Schiff base protonation in wild-type and mutants, it would be good to present detailed spectroscopic data.

Detailed spectroscopic characterization of wild-type *Gt*ACR1 and several of its mutants has already been presented in two PNAS articles (Sineshchekov et al., 2015 and 2016) referred to in this study. We have added with discussion in the revision two new figures (Figure 4—figure supplement 1 and Figure 5—figure supplement 1) showing the action spectra of photocurrents generated by the F160G, Y72F and Y207F mutants.

c) The authors should include a discussion paragraph highlighting key unique features that distinguish their work from the work of Kim et al., (2018).

We thank the Editors for the opportunity to add this informative section. We have added a subsection ““Comparison with the recently published *Gt*ACR1 structure” at the end of Results and Discussion section. We believe that overall the independent analyses of the structures by both our groups provide complimentary information creating a firm basis for elucidating the functional mechanism and for further optimization of *Gt*ACR1 as a potent neuron-silencing optogenetic tool.